# Nero: A Deterministic Leaderless Consensus Algorithm for DAG-Based Cryptocurrencies

Rui Morais [1,*] , Paul Crocker [1,2,*] and Valderi Leithardt [3,4]

1 Department of Informatics, University of Beira Interior, 6201-001 Covilhã, Portugal
2 Instituto de Telecomunicações and Department of Informatics, University of Beira Interior, 6201-001 Covilhã, Portugal
3 VALORIZA—Research Centre for Endogenous Resource Valorization, Polytechnic Institute of Portalegre, 7300-110 Portalegre, Portugal
4 COPELABS, Lusófona University of Humanities and Technologies, Campo Grande 376, 1749-024 Lisboa, Portugal
* Correspondence: ru.morais@ubi.pt (R.M.); crockercaria@gmail.com (P.C.)

**Abstract:** This paper presents the research undertaken with the goal of designing a consensus algorithm for cryptocurrencies with less latency than the current state-of-the-art while maintaining a level of throughput and scalability sufficient for real-world payments. The result is Nero, a new deterministic leaderless byzantine consensus algorithm in the partially synchronous model that is especially suited for Directed Acyclic Graph (DAG)-based cryptocurrencies. In fact, Nero has a communication complexity of $O(n^3)$ and terminates in two message delays in the good case (when there is synchrony). The algorithm is shown to be correct, and we also show that it can provide eventual order. Finally, some performance results are given based on a proof of concept implementation in the Rust language.

**Keywords:** consensus; byzantine; directed acyclic graph





## 1. Introduction

The consensus problem was first introduced in [1]. It can be described as a set of entities that wish to agree on something, which could be a value, an action or a statement. At first glance, it appears a simple problem to solve, especially if all parties are honest. However, it becomes complex when some subset of those entities can behave unexpectedly, such as by not participating and leaving the consensus process without revealing their choice to the other parties or by acting maliciously by actively trying to sabotage the consensus scheme so that no consensus is reached or such that honest parties decide differently instead of arriving at a consensus.

This problem is typically studied in the context of computer science, where the entities are processes on computers and where the computers communicate with each other through an unreliable network such as the internet. However, this problem and the underlying principles can be applied to other areas, such as politics and game theory. In fact, one of the seminal works in this area is a metaphor for a group of generals on the battlefield who need to decide if they attack or not [2], and this is where the term "byzantine" comes from, meaning any type of faulty behavior, also called "arbitrary" faults. Ever since, it has become one of the most studied problems in computer science due to its both theoretical and practical importance. From a theoretical standpoint, it has been shown to be equivalent to other problems, such as atomic broadcast and state machine replication, meaning that if you can solve one, you can also solve the others. From a practical standpoint, it is very useful in the deployment of real-world distributed systems.

It started with leaderless synchronous consensus algorithms [2,3], but these had limited application in practice since distributed systems in the real world are not synchronous.

This has led to the study of asynchronous distributed systems and to one of the most important theorems in the field: the FLP impossibility theorem [4]. According to this theorem, it is not possible to have agreement and termination in the asynchronous model if one of the processes crash. Throughout the years, there have been many ways to circumvent this, such as randomization [5], modifying the model [6] or the properties [7] of the problem and byzantine fault detectors [8,9].

Despite being an important field of research in computer science, distributed systems and particularly consensus algorithms gained new momentum with the emergence of Bitcoin [10], the first decentralized-permissionless network that uses a blockchain as a ledger to store the transactions. Its consensus mechanism, the Nakamoto consensus [11], led to the development of a whole new family of consensus algorithms based on Proof of Work (PoW).

However, soon the limitations of a design based on PoW became evident, such as the low throughput and high latency, and new alternatives started to appear, such as Proof of Stake (PoS) [12–14] and PoS with a ledger structure based on directed graphs instead of a linear, chronological order.

Despite improving on the limitations of Bitcoin, new alternatives based on proof of stake also solved state machine replication [15] by sequentially executing consensus instances for agreeing on each block of transactions to append to the blockchain.

In order to improve scalability, another ledger structure called directed acyclic graph has gained popularity. This allows for consensus instances to be executed concurrently instead of sequentially and thereby improves the latency and throughput of the system [16]. Examples are [17–20].

A DAG is a graph that is made up of a set of vertices (or nodes) connected by directed edges (or arcs) such that a closed chain is not possible. A blockchain is a type of acyclic graph where each vertex is a block of transactions and is connected to only another vertex sequentially. In a general DAG, each vertex can be connected to many vertices, and these connections are not made sequentially but concurrently instead. A blockchain can be seen as a DAG where only one edge can be connected to a vertex.

In the context of cryptocurrencies, this allows for a greater throughput and less latency when validating and adding transactions to the ledger compared to a blockchain. However, the downside is that it is much more difficult to order those transactions and, consequently, for the nodes of the network to have a common sense of time. A common sense of time is useful, for example, for synchronous network upgrades, for pruning the ledger and bootstrapping. Another problem is scalability and decentralization, which are connected to each other.

Many consensus algorithms are also used to assign a node the status of the leader. However, a leader-based approach has limitations, especially when nodes are geographically dispersed over a wide area network. Firstly, the latency for clients far from the leader is obviously increased. Furthermore, the leader can become a bottleneck or its network performance may degrade, leading to an overall decrease in system-wide performance decreases. Finally, if the leader fails, the whole system cannot serve new requests until an election of a new leader takes place, thereby affecting availability.

Therefore, a ledger based on a DAG design is most suited for leaderless consensus algorithms, which can be probabilistic [21,22] or deterministic [23]. In fact, Leaderless State Machine Replication (SMR) offers appealing properties with respect to leader-driven approaches. Protocols are faster in the best case, suffer from no downtime when the leader fails, and distribute the load among participants.

In this paper, we present a new leaderless deterministic consensus algorithm and a mechanism that provides eventual order, making it suitable for DAG-based cryptocurrencies. Some theoretical properties of the algorithms are given, such as a complexity analysis and correctness proprieties. The final contribution is an open-source implementation in the Rust language.

In Section 2, some preliminaries are described, in particular, multi-valued leaderless consensus properties are defined, and the overall system model is defined. In Section 3, details of the new Nero consensus algorithm are given. In Section 4, the correctness of the algorithm is discussed, and performance results from a proof of concept implementation in Rust are given. Finally, in the last section, a discussion and final conclusions are made.

## 2. Preliminares

### 2.1. Multi-Valued Consensus

We use a variant of the multi-valued byzantine consensus problem called Validity Predicate-based Byzantine consensus [24], with the following properties.

- Agreement: No two correct processes decide on different values.
- Termination: All correct processes eventually decide on a value.
- Validity: A decided value is valid, i.e., it satisfies the predefined predicate denoted valid().

**Correctness.** An algorithm satisfies the Multi-valued Consensus if it satisfies validity, agreement and termination.

### 2.2. System Model

The system consists of a set $P$ of n asynchronous processes, namely $P = p_1, \ldots, p_n$, where asynchronous means that each process proceeds at its own speed, which can vary with time and remains unknown to the other processes. Up to $f$ processes among $n = 3f + 1$ can fail: at most, one is suspended [25], and $f - 1$ can behave arbitrarily or be Byzantine.

We use a variant of the partially synchronous system model [6] where the system can be in one of two states in a given time: GST (Global Stabilization Time) and non-GST. In the first, there is a known bound $\Delta$ to all sent messages, meaning that a message broadcasted by a correct process at time $t$ will be delivered by all correct processes before $t + \Delta$. In a non-GST state, this is not guaranteed to happen; however, it is assumed that the system reaches GST at least once.

## 3. The Nero Algorithm

In this section, we first give an overview and intuition behind the design of the Nero algorithm and describe the data structures necessary for the execution of the algorithm. Then we present the algorithm, prove its correctness and finally describe the mechanism to satisfy the eventual order.

### 3.1. Overview and Intuition

Our goal is to develop a consensus algorithm that achieves consensus for a cryptocurrency with a DAG ledger instead of a blockchain. Typically, in the latter, there is an algorithm for choosing the leader of the round, who proposes a block of transactions to be validated, and the other processes are also validated or not (binary consensus). In a blockchain, a non-malicious double spend attempt can happen in the selection of the leader when the algorithm selects more than one leader for the same round who proposes different blocks of transactions (this can happen in Bitcoin if more than two miners find the solution of the proof of work more or less at the same time, although it is very unlikely due to the high latency of more or less 10 min). However, if only one leader is chosen in a given round, a double spend attempt can only happen intentionally if the leader includes conflicting transactions in the same block.

With a DAG structure, it is different because there is no leader in each round, so every process can propose transactions and those transactions can be conflicted, i.e., have the same origin block, which is called a double-spend attempt. Since the latency is very low, different correct processes can receive different conflicting transactions in a different order and make a conflicting vote. Because the probability of double spending attempts happening in an unintentional way is higher than in a blockchain, the amount of computational resources needed to resolve them is also higher. Because of that, and assuming that conflicting

transactions (not votes) can only happen in a malicious way or due to a programming error, our consensus algorithm discards all of them and votes nil instead of choosing a winner transaction to be validated from the set of conflicting transactions.

Similarly, if a correct process receives at least two conflicting transactions, it votes nil on that election. If it has only received one transaction but at least $f + 1$ processes vote nil, it means that at least one correct process received a double spend attempt, and so it also votes nil. This way, the amount of computational resources to solve a double spend attempt, or fork, will be lower, and the latency of the consensus algorithm will also be lower.

Now we need to take into account the processes that vote in a malicious way, so we need a way to validate messages. One way to do that is to append a proof to each message with the messages of the previous round in which it is based; however, this has a very high communication complexity. Another way is to send only the hashes of the messages as proof. If the receiving process does not have the corresponding messages to the hashes yet, it considers the message pending and waits for the previous messages to validate it. This reduces the communication complexity; however, we can still reduce it more and not send any proof at all. A validation algorithm checks if the received message is valid, pending or invalid according to the received messages of the previous round.

### 3.2. Data Structures

We abstract the data structures and refer to only the contents relevant to the operation of the consensus algorithm.

**Block.** Data containing the value transferred from the sender account to the receiver account.

**Transaction.** To simplify the presentation, we abstract the contents of a transaction that are not relevant to the consensus algorithm, and we assume that a transaction is composed of the origin block hash and the new block hash.

**Round.** A simple integer that starts at 0 and ends in the round where the value of an election is decided.

**Elections.** Each correct process maintains a hashmap of the active elections, where the key is a block hash (corresponding to the origin block of a transaction) and the value is the state of the election. An election can have multiple competing transactions (called forks) if they have the same origin block and the purpose of the consensus is that all correct processes validate one of the transactions or none.

**Election State.** The state of an election is represented by a hashmap, where the key is the election hash and the value a set of round states.

**Round State.** The state of a round is represented by a hashmap, where the key is the round number and the value the tally of the messages of that round. It also contains the validated and pending messages.

**Tally.** Each correct process computes a tally of the messages received in each round, with the total number of votes or commits in each value (block hash or nil).

**Messages.** A message is composed of a:

- Type: VOTE or COMMIT,
- Value: a block hash or a nil value,
- Round.

A message $m$ can be valid, pending or invalid. A message $m$ of round $r$ is considered valid by a correct process $p$ if there is a set of at least $2f + 1$ messages received by $p$ in the previous round $r − 1$ that are compatible with the value and type of message $m$. A message is considered pending if there is no set compatible, but it is still possible to have that set by receiving the remaining messages of round $r − 1$. It is considered invalid if it is not possible to have a compatible set with the message.

**Timer.** At the start of each round, a correct process starts a timer $t_r$, where $r$ is the round.

**New round.** A correct process starts a new round $r + 1$ when it has received at least $2f + 1$ messages in round $r$ and the timer $t_r$ has expired. During GST, all messages from

correct processes will be received before the timer expires; however, a correct process has no way of knowing that it is in GST, i.e., if it receives $2f + 1$ messages and the timer expires, it is not guaranteed that those messages are all from correct processes.

### 3.3. The Algorithm

**Validation rules.** After receiving a message $m$ of round $r$, a correct process validates it and decides one of the following statuses:

- Valid if there is a set of at least $2f + 1$ messages of round $r - 1$ that are compatible with the value and type of $m$.
- Pending if there is not a set of at least $2f + 1$ messages of round $r - 1$ that are compatible with the value and type of $m$ but it is still possible to receive new messages of round $r - 1$ to produce that set.
- Invalid if there is not a set of at least $2f + 1$ messages of round $r - 1$ that are compatible with the value and type of m, and it is not possible to receive new messages of round $r - 1$ to produce that set.

**Decision rules.** With the start of a new round $r + 1$, a new decision has to be made to decide the message that the process is going to send in the new round. This message is based on the messages received in the previous round $r$ (at least $2f + 1$ valid messages in total), in the following way (only one rule can be applied):

- If process $p$ has received at least $2f + 1$ valid messages of type COMMIT with the same value, decide that value. There is no need to send a new message because all the other correct processes will eventually receive the same messages and decide the same value as well.
- Else if process $p$ has received at least one valid message of type COMMIT with some value $v$, send a message of type COMMIT with value $v$. (There cannot be two commits of different values in the same round).
- Else if process $p$ has received at least $2f + 1$ valid messages of type VOTE with the same value, send a message of type COMMIT with that value.
- Else if process $p$ has received at least one valid message of type COMMIT with some value $v$, send a message of type COMMIT with that value.
- Else if process $p$ has received at least one valid message of type COMMIT with some value $v$ in round $r$ and has already committed to another value $v'$ in round $r'$, send a message of type COMMIT with value $v$ only if round $r > r'$. If $r < r'$, send a message of type *COMMIT* with the previous committed value $v'$.
- Else if process $p$ has received at least $f + 1$ valid messages of type VOTE with value nil, send a message of type VOTE with value nil.
- If no block hash value has at least $f + 1$ votes, send a message of type VOTE with value nil.
- Else if process $p$ has not received at least $f + 1$ valid messages of type VOTE with value nil, send a message of type VOTE with the value with most votes.

**Execution.** In each round $r$, a correct process waits for at least $2f + 1$ valid messages and for the timer $t_r$ to expire. Every time it receives a valid message, it does the following:

- Inserts it in the respective election state.
- Updates the tally of the election.
- Tries to validate previous pending messages based on the validation rules.
- Broadcasts $m$ to the network.
- Computes a new message $m'$ of round $r + 1$ based on the received messages and the decision rules.
- Broadcasts the message $m'$ to the other processes.

### 3.4. Correctness

In this section, we discuss the correctness of the Nero consensus algorithm.

**Lemma 1.** *There cannot be two or more valid commits with different values in a given round.*

**Proof.** Since, per the consensus algorithm, a correct process never votes or commits more than one block hash in the same round in the same election, it is not possible to have two sets of $2f + 1$ of different values without at least one correct process voting for two different values, so we prove by contradiction. □

**Lemma 2.** *If a correct process commits a value $v$ in round $r$ and it receives a valid commit of another value $v'$ in round $r'$, $r' > r$, no correct process could have decided $v$ in round $r$.*

**Proof.** If a correct process decides value $v$ in round $r$, it means that it has received at least $2f + 1$ commits with value $v$ in round $r - 1$. Since at least $f + 1$ of those commits are from correct processes, they will not change its value and so the $2f + 1$ valid messages of value $v'$ of type VOTE needed to commit value $v'$ will not be produced. Inversely, if there was a commit of value $v'$, it means that value $v$ was not decided in a previous round. □

**Lemma 3.** *The Nero algorithm satisfies Termination.*

**Proof.** We need to prove that all correct processes decide. For that, we have to prove that all correct processes eventually commit a value and then that all correct processes eventually decide a value. The proof follows from Lemmas 4 and 5. □

**Lemma 4.** *All correct processes eventually commit a value.*

**Proof.** First, we prove that at least one correct process eventually commits. Since a byzantine process can send different valid votes (with different values) to different correct processes in a given round, it is possible that no correct process commits during the non-GST phase.

However, during the GST, all correct processes will eventually vote the same value in a given round $r$ because they will all receive the same votes in round $r - 1$. □

**Lemma 5.** *All correct processes eventually decide a value.*

**Proof.** First, we prove that at least one correct process eventually decides, which happens because all correct processes will eventually commit the same value.

During the GST, all correct processes will eventually decide as well since they will receive the valid commits in which the correct process based its decided value. □

**Lemma 6.** *The Nero algorithm satisfies Agreement.*

**Proof.** We need to prove that all correct processes decide the same value. Since we already proved in Lemma 5 that all correct processes eventually decide, we need to prove that they all decide the same value.

Suppose that a correct process $p$ decides value $v$ in round $r$. This means that it has received at least $2f + 1$ commits with value $v$ in round $r - 1$. At most there are $f$ byzantine processes, so at least $f + 1$ of those commits are from correct processes, which means that they will only change their commit if they receive a valid commit with value $v'$ in a round $r', r' > r - 1$.

If $r' = r$, this cannot happen due to Lemma 1. If $r' > r$, for a valid commit with value $v'$ to happen, there needs to be at least $2f + 1$ valid votes in a previous round $r' - 1$, which is not possible since at least $f + 1$ correct processes will not change their commit with value $v$ in round $r$ and vote with value $v'$ since $r' - 1 >= r$. Therefore, we prove by contradiction.

Lemma 2 ensures that a correct process can change its commit to a subsequent valid commit while maintaining safety. □

**Lemma 7.** *The Nero algorithm satisfies Validity.*

**Proof.** A correct process only sends messages with a valid value so only valid values can be decided. □

**Theorem 1.** *The Nero algorithm achieves a multi-valued consensus, i.e., Termination, Validation and Agreement.*

**Proof.** Follows from Lemmas 3, 6 and 7. □

*3.5. Eventual State Machine Replication*

In this section, we adapt the Nero consensus algorithm to achieve a variant of state machine replication suited for DAG-based systems called eventual state machine replication. Concretely, we change the definition of replica coordination of [15] to all non-faulty replicas *eventually* receive and process the same sequence of requests.

This property can be decomposed into two parts, Agreement and Eventual Order: Agreement requires all (non-faulty) replicas to receive all requests, and Eventual Order requires that the order of received requests is eventually the same at all replicas.

In this section, we propose a way to eventually order the transactions while maintaining the properties of the consensus algorithm. We do this by modifying the contents of the messages sent (in this case, the transactions), with the client adding a timestamp to it, and modifying the valid() predicate accordingly. In addition to the previous validation conditions, the node now has to also validate the timestamp of the transaction according to its own local timestamp when it receives it in order to discard invalid timestamps. Concretely, a timestamp t of a transaction is considered valid if $l - d < t < l + d$, where $l$ is the local timestamp and $d$ is the value to account for the local timestamps desynchronization of the different nodes and the delay of broadcasting a message through the network. The local timestamps can be synchronized using a decentralized protocol such as Network Time Protocol (NTP) [26] or Network Time Security (NTS) [27].

This is not meant to provide an accurate timestamp of when a given transaction happened but only provide a global ordering of the ledger and a common sense of time to the nodes of the network. There is a probability of valid transactions being discarded when the network is not on GST, but they can always be resubmitted with a new timestamp. In order to minimize this, a node can vote on a transaction with an "invalid" timestamp (in its view) if at least $f + 1$ nodes voted on that transaction, meaning that at least one correct process received the transaction in a timely manner.

The ordering of the messages is then performed by its timestamp first and, if there is more than one message, with the same timestamp, by the hash of the message using a hash with the properties of [28]: compression, one way, weak collision resistance and strong collision resistance.

Proof. Assuming the agreement and termination properties of the underlying consensus algorithm, all correct processes will eventually decide on the same set of values. In the case that the values all have different timestamps, the order is naturally done. In the case that different values have the same timestamp, the hash of those values assures a unique deterministic order because any value is hashable (compression property), and each hash is unique in practice (weak collision resistance property).

**4. Results**

In this section, we compare the Nero algorithm to other related work from a theoretical standpoint (communication complexity and message delays) and analyze its implementation and use case in practice.

### 4.1. Comparison with Related Work

In Table 1, we compare other relevant deterministic consensus algorithms using different parameters, such as the message complexity during normal case and view change, the latency (message delays) and if they are leaderless or not. Note that for completeness we also add to the table the parameters from the new Nero consensus algorithm, which will be described in Section 4. A more complete comparison of consensus layer techniques for several algorithms and indeed other parameters can be found in [29].

**Table 1.** Comparison of different consensus algorithms.

|  | Normal Case | View Change | Message Delays | Leaderless |
|---|---|---|---|---|
| **PBFT** [30] | $O(n^3)$ | $O(n^4)$ | 3 | No |
| **HotStuff** [31] | $O(n^2)$ | $O(n^2)$ | 8 | No |
| **IBFT** [32] | $O(n^2)$ | $O(n^2)$ | 3 | No |
| **Tendermint** [33] | $O(n^3)$ | - | 3 | No |
| **DBFT** [34] | $O(n^3)$ | - | 4 | No |
| **Archipelago** [23] | $O(n^3)$ | - | 5 | Yes |
| **Nero** | $O(n^3)$ | - | 2 | Yes |

Tendermint [33] is an improved version of PBFT that does not need a view change protocol but still has a greater latency due to the PROPOSAL message of the leader, which Nero does not have since every process makes its proposal a PREVOTE message.

IBFT [32] has a better communication complexity but has a view change, which adds to the implementation complexity, and it is not leaderless.

DBFT [34] does not require signatures and has the same communication complexity; however, it is not leaderless according to our definition due to its requirement of a weak coordinator and a greater latency.

HotStuff, despite having a better communication complexity, has high latency and its throughput drops to zero when the leader fails and until some view-change completes [35].

Archipelago [23] is the only other leaderless deterministic consensus algorithm and has the same communication complexity; however, it has a greater latency.

### 4.2. Implementation

We implemented the Nero consensus algorithm in the Rust programming language (https://github.com/Fiono11/nero_consensus.git (accessed on 3 January 2023), and we benchmarked its latency and throughput for a different number of processes in Ubuntu 18.04, Intel Core i7-4790 3.60 GHz, 16 GB RAM. As expected, the latency increases and the throughput decreases with the number of nodes, as shown in Figure 1. We show that the algorithm has very little latency, which was the primary goal, and that its throughput is enough to have real-world applications, although it can still be improved with optimizations such as batching of votes.

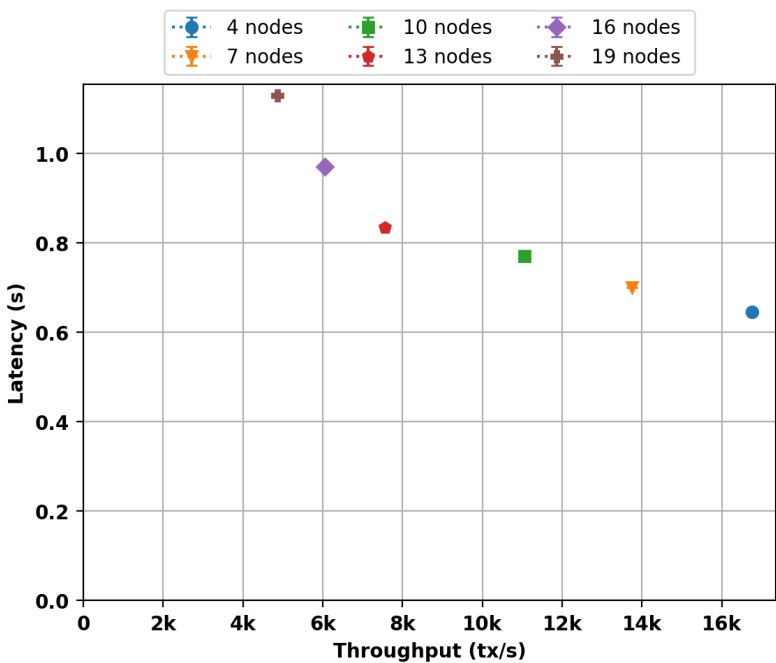

**Figure 1.** Latency and throughput of Nero.

## 5. Conclusions

This work presents a deterministic leaderless consensus and provides a mechanism of eventual ordering that makes it suitable for DAG-based cryptocurrencies without harming scalability and decentralization.

We show that it is competitive with other consensus algorithms in the communication complexity and improves the latency. The prototype implementation indicates that Nero can be used in real world payment applications; however, it can still be optimized and improved.

Future work will focus on formal verification using ByMC [36] and testing the implementation in a real-world scenario.

**Author Contributions:** Conceptualization, R.M.; Methodology, V.L.; Investigation, P.C.; Writing—original draft, R.M.; Writing—review & editing, R.M., P.C. and V.L.; Supervision, P.C. All authors have read and agreed to the published version of the manuscript.

**Funding:** This work is funded by FCT/MCTES through national funds and when applicable co-funded EU funds under the project UIDB/EEA/50008/2020 and by NOVA LINCS (UIDB/04516/2020) with the financial support of FCT—Fundação para a Ciência e a Tecnologia, by operation Centro 2020—Centro-01-0145-FEDER-000019-C4—Centro de Competências em Cloud Computing and by the project UIDB/05064/2020 (VALORIZA—Research Centre for Endogenous Resource Valorization).

**Data Availability Statement:** Not applicable.

**Acknowledgments:** The authors would like to thank the support of the RELEASE, RELiablE And SEcure Computation Research Group at the University of Beira Interior and the Networks Architectures and Protocols group of the Instituto de Telecomunicações.

**Conflicts of Interest:** The authors declare no conflict of interest.

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
