# Peer review of "Nero: A Deterministic Leaderless Consensus Algorithm for DAG-Based Cryptocurrencies"

_algorithms, doi:10.3390/a16010038_

Round 1

Reviewer 1 Report

The authors propose a leaderless deterministic consensus algorithm, and the the subject is topical.  However, to make this paper publishable, the authors need to respond to the following concerns:

(1)The Nero algorithm is unclear, and it would be helpful to incorporate a better explanation, preferably using a simple example or figure .

(2)The research contribution in this paper must be made clear, new algorithm or  improved algorithm followed by[23]?

(3)The literature reviews are not properly done in this paper. It's also important if the author can summarize the advantages, and disadvantages of each strategy in a table. 

(4)Complexity analysis from section 3 must be reviewed. The experiment is not comprehensive and thorough. For example, what is the detail configuration of the proposed comparative experiments? 

Author Response

We improved on the results with a benchmark of an open source implementation of the consensus algorithm and we described the new algorithm in a simpler way. The advantages and disadvantages of blockchain vs DAG and leaderless vs leader based algorithms are stated.

Reviewer 2 Report

1. The abstract must be completely rewritten. It must refer to the research carried out and the results obtained. 2. The results do not highlight the scientific nature of the work. They MUST be detailed and explained for each stage in this study. Without detailing them, the work has no scientific value. 3. Conclusions are almost non-existent. The authors must detail with scientific evidence, the achievements, areas of application, and the ways to follow. The limitations and constraints of this study must also be pointed out.

Author Response

We improved on the results with a benchmark of an open source implementation of the consensus algorithm.

We added a section explaining the intuition behind the algorithm and its design choices.

The abstract and conclusions were rewritten.

Round 2

Reviewer 2 Report

The manuscript has been sufficiently improved, according to the recommendations.